# Scalable kernels for graphs with continuous attributes

**Aasa Feragen, Niklas Kasenburg**
Machine Learning and Computational Biology Group
Max Planck Institutes Tübingen and DIKU, University of Copenhagen
{aasa,niklas.kasenburg}@diku.dk

**Jens Petersen[1],**
**Marleen de Bruijne[1,2]**
[1]DIKU, University of Copenhagen
[2] Erasmus Medical Center Rotterdam
{phup,marleen}@diku.dk

**Karsten Borgwardt**
Machine Learning and Computational Biology Group
Max Planck Institutes Tübingen
Eberhard Karls Universität Tübingen
karsten.borgwardt@tuebingen.mpg.de

## Abstract

While graphs with continuous node attributes arise in many applications, state-of-the-art graph kernels for comparing continuous-attributed graphs suffer from a high runtime complexity. For instance, the popular shortest path kernel scales as $\mathcal{O}(n^4)$, where $n$ is the number of nodes. In this paper, we present a class of graph kernels with computational complexity $\mathcal{O}(n^2(m + \log n + \delta^2 + d))$, where $\delta$ is the graph diameter, $m$ is the number of edges, and $d$ is the dimension of the node attributes. Due to the sparsity and small diameter of real-world graphs, these kernels typically scale comfortably to large graphs. In our experiments, the presented kernels outperform state-of-the-art kernels in terms of speed and accuracy on classification benchmark datasets.

## 1 Introduction

Graph-structured data appears in many application domains of machine learning, reaching from Social Network Analysis to Computational Biology. Comparing graphs to each other is a fundamental problem in learning on graphs, and graph kernels have become an efficient and widely-used method for measuring similarity between graphs. Highly scalable graph kernels have been proposed for graphs with thousands and millions of nodes, both for graphs without node labels [1] and for graphs with discrete node labels [2]. Such graphs appear naturally in applications such as natural language processing, chemoinformatics and bioinformatics. For applications in medical image analysis, computer vision or even bioinformatics, however, continuous-valued physical measurements such as shape, relative position or other measured node properties are often important features for classification. An open challenge, which is receiving increased attention, is *to develop a scalable kernel on graphs with continuous-valued node attributes*.

We present the GraphHopper kernel between graphs with real-valued edge lengths and any type of node attribute, including vectors. This kernel is a convolution kernel counting sub-path similarities. The computational complexity of this kernel is $\mathcal{O}(n^2(m + \log n + \delta^2 + d))$, where $n$ and $m$ are the number of nodes and edges, respectively; $\delta$ is the graph diameter; and $d$ is the dimension of the node attributes. Although $\delta = n$ or $m = n^2$ in the worst case, this is rarely the case in real-world graphs, as is also illustrated by our experiments. We find empirically in Section 3.1 that our GraphHopper kernel tends to scale quadratically with the number of nodes on real data.

## 1.1 Related work

Many popular kernels for structured data are sums of substructure kernels:

$$k(G, G') = \sum_{s \in \mathscr{S}} \sum_{s' \in \mathscr{S}'} k_{\mathrm{sub}}(s, s').$$

Here $G$ and $G'$ are structured data objects such as strings, trees and graphs with classes $\mathscr{S}$ and $\mathscr{S}'$ of substructures, and $k_{\mathrm{sub}}$ is a substructure kernel. Such $k$ are instances of *R-convolution kernels* [3].

A large variety of kernels exist for structures such as strings [4, 5], finite state transducers [6] and trees [5, 7]. For graphs in general, kernels can be sorted into categories based on the types of attributes they can handle. The *graphlet kernel* [1] compares unlabeled graphs, whereas several kernels allow node labels from a finite alphabet [2, 8]. While most kernels have a runtime that is at least $\mathcal{O}(n^3)$, the *Weisfeiler-Lehman kernel* [2] uses efficient sorting, hashing and counting algorithms that take advantage of repeated occurrences of node labels from the finite label alphabet, and achieves a runtime which is at most quadratic in the number of nodes. Unfortunately, this does not generalize to graphs with vector-valued node attributes, which are typically all distinct samples from an infinite alphabet.

The first kernel to take advantage of non-discrete node labels was the *random walk kernel* [9–11]. It incorporates edge probabilities and geometric node attributes [12], but suffers from tottering [13] and is empirically slow. Kriege *et al.* [14] adopt the idea of comparing matched subgraphs, including vector-valued attributes on nodes and edges. However, this kernel has a high computational and memory cost, as we will see in Section 3. Other kernels handling non-discrete attributes use edit-distance and subtree enumeration [15]. While none of these kernels scale well to large graphs, the *propagation kernel* [16] is fast asymptotically and empirically. It translates the problem of continuous-valued attributes to a problem of discrete-valued labels by hashing node attributes. Nevertheless, its performance depends strongly on the hashing function and in our experiments it is outperformed in classification accuracy by kernels which do not discretize the attributes.

In problems where continuous-valued node attributes and inter-node distance $d_G(v, w)$ along the graph $G$ are important features, the *shortest path kernel* [17], defined as

$$k_{SP}(G, G') = \sum_{v, w \in V} \sum_{v', w' \in V'} k_n(v, v') \cdot k_l \left( d_G(v, w), d_{G'}(v', w') \right) \cdot k_n(w, w'),$$

performs well in classification. In particular, $k_{SP}$ allows the user to choose any kernels $k_n$ and $k_l$ on nodes and shortest path length. However, the asymptotic runtime of $k_{SP}$ is generally $\mathcal{O}(n^4)$, which makes it unfeasible for many real-world applications.

## 1.2 Our contribution

In this paper we present a kernel which also compares shortest paths between node pairs from the two graphs, but with a different path kernel. Instead of comparing paths via products of kernels on their lengths and endpoints, we compare paths through kernels on the nodes encountered while "hopping" along shortest paths. This particular path kernel allows us to decompose the graph kernel as a weighted sum of node kernels, initially suggesting a potential runtime as low as $\mathcal{O}(n^2 d)$. The graph structure is encoded in the node kernel weights, and the main algorithmic challenge becomes to efficiently compute these weights. This is a combinatorial problem, which we solve with complexity $\mathcal{O}(n^2(m + \log n + \delta^2))$. Note, moreover, that the GraphHopper kernel is parameter-free except for the choice of node kernels.

**The paper is organized as follows.** In Section 2 we give short formal definitions and proceed to defining our kernel and investigating its computational properties. Section 3 presents experimental classification results on different datasets in comparison to state-of-the-art kernels as well as empirical runtime studies, before we conclude with a discussion of our findings in Section 4.

## 2 Graphs, paths and GraphHoppers

We shall compare undirected graphs $G = (V, E)$ with edge lengths $l \colon E \to \mathbb{R}_+$ and node attributes $A \colon V \to X$ from a set $X$, which can be any set with a kernel $k_n$; in our data $X = \mathbb{R}^d$. Denote

$n = |V|$ and $m = |E|$. A subtree $T \subset G$ is a subgraph of $G$ which is a tree. Such subtrees inherit node attributes and edge lengths from $G$ by restricting the attribute and length maps $A$ and $l$ to the new node and edge sets, respectively. For a tree $T = (V, E, r)$ with a root node $r$, let $p(v)$ and $c(v)$ denote the parent and the children of any $v \in V$.

Given nodes $v_a, v_b \in V$, a path $\pi$ from $v_a$ to $v_b$ in $G$ is defined as a sequence of nodes

$$\pi = [v_1, v_2, v_3, \ldots, v_n],$$

where $v_1 = v_a$, $v_n = v_b$ and $[v_i, v_{i+1}] \in E$ for all $i = 1, \ldots, n-1$. Let $\pi(i) = v_i$ denote the $i^{th}$ node encountered when "hopping" along the path. Given paths $\pi$ and $\pi'$ from $v$ to $w$ and from $w$ to $u$, respectively, let $[\pi, \pi']$ denote their composition, which is a path from $v$ to $u$. Denote by $l(\pi)$ the *weighted length* of $\pi$, given by the sum of lengths $l(v_i, v_{i+1})$ of edges traversed along the path, and denote by $|\pi|$ the *discrete length* of $\pi$, defined as the number of nodes in $\pi$. The *shortest path* $\pi_{ab}$ from $v_a$ to $v_b$ is defined in terms of weighted length; if no edge length function is given, set $l(e) = 1$ for all $e \in E$ as default. The *diameter* $\delta(G)$ of $G$ is the maximal number of nodes in a shortest path in $G$, with respect to weighted path length.

In the next few lemmas we shall prove that for a fixed a source node $v \in V$, the directed edges along shortest paths from $v$ to other nodes of $G$ form a well-defined directed acyclic graph (DAG), that is, a directed graph with no cycles.

First of all, subpaths of shortest paths $\pi_{vw}$ with source node $v$ are shortest paths as well:

**Lemma 1.** [18, Lemma 24.1] *If $\pi_{1n} = [v_1, \ldots, v_n]$ is a shortest path from $v_1 = v$ to $v_n$, then the path $\pi_{1n}(1 \colon i)$ consisting of the first $i$ nodes of $\pi_{1n}$ is a shortest path from $v_1 = v$ to $v_i$.* $\qquad\square$

Given a source node $v \in G$, construct the directed graph $G_v = (V_v, E_v)$ consisting of all nodes $V_v$ from the connected component of $v$ in $G$ and the set $E_v$ of all directed edges found in any shortest path from $v$ to any given node $w$ in $G_v$. Any directed walk from $v$ in $G_v$ is a shortest path in $G$:

**Lemma 2** *If $\pi_{1n}$ is a shortest path from $v_1 = v$ to $v_n$ and $(v_n, v_{n+1}) \in E_v$, then $[\pi_{1n}, [v_n, v_{n+1}]]$ is a shortest path from $v_1 = v$ to $v_{n+1}$.*

**Proof.** Since $(v_n, v_{n+1}) \in E_v$, there is a shortest path $\pi_{1(n+1)} = [v_1, \ldots, v_n, v_{n+1}]$ from $v_1 = v$ to $v_{n+1}$. If this path is shorter than $[\pi_{1n}, [v_n, v_{n+1}]]$, then $\pi_{1(n+1)}(1 : n)$ is a shortest path from $v_1 = v$ to $v_n$ by Lemma 1, and it must be shorter than $\pi_{1n}$. This is impossible, since $\pi_{1n}$ is a shortest path.$\square$

**Proposition 3** *The shortest path graph $G_v$ is a DAG.*

**Proof.** Assume, on the contrary, that $G_v$ contains a cycle $c = [v_1, \ldots, v_n]$ where $(v_i, v_{i+1}) \in E_v$ for each $i = 1, \ldots, n-1$ and $v_1 = v_n$. Let $\pi_{v1}$ be the shortest path from $v$ to $v_1$. Using Lemma 2 repeatedly, we see that the path $[\pi_{v1}, c]$ is a shortest path from $v$ to $v_n = v_1$, which is impossible since the new path must be longer than the shortest path $\pi_{v1}$. $\qquad\square$

## 2.1 The GraphHopper kernel

We define the GraphHopper kernel as a sum of path kernels $k_p$ over the families $\mathscr{P}, \mathscr{P}'$ of shortest paths in $G, G'$:

$$k(G, G') = \sum_{\pi \in \mathscr{P}, \pi' \in \mathscr{P}'} k_p(\pi, \pi'),$$

In this paper, the path kernel $k_p(\pi, \pi')$ is a sum of node kernels $k_n$ on nodes simultaneously encountered while simultaneously hopping along paths $\pi$ and $\pi'$ of equal discrete length, that is:

$$k_p(\pi, \pi') = \begin{cases} \sum_{j=1}^{|\pi|} k_n\left(\pi(j), \pi'(j)\right) & \text{if } |\pi| = |\pi'|, \\ 0 & \text{otherwise.} \end{cases} \tag{4}$$

It is clear from the definition that $k(G, G')$ decomposes as a sum of node kernels:

$$k(G, G') = \sum_{v \in V} \sum_{v' \in V'} w(v, v') k_n(v, v'), \tag{5}$$

where $w(v, v')$ counts the number of times $v$ and $v'$ appear at the same hop, or coordinate, $i$ of shortest paths $\pi, \pi'$ of equal discrete length $|\pi| = |\pi'|$. We can decompose the weight $w(v, v')$ as

$$w(v, v') = \sum_{j=1}^{\delta} \sum_{i=1}^{\delta} \sharp\left\{(\pi, \pi') | \pi(i) = v, \pi'(i) = v', |\pi| = |\pi'| = j\right\} = \langle M(v), M(v') \rangle,$$

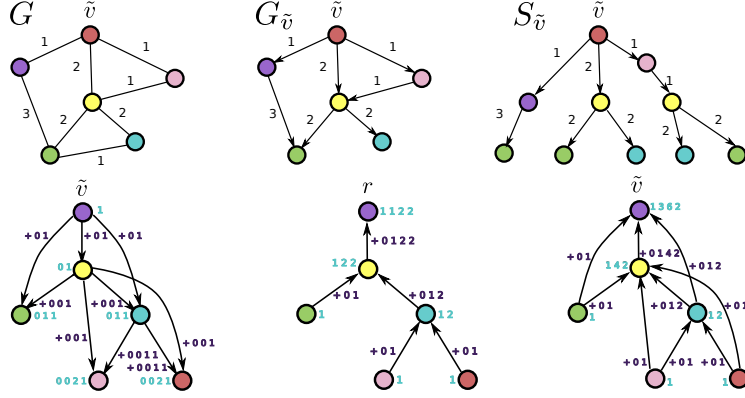

Figure 1: **Top:** Expansion from the graph $G$, to the DAG $G_{\tilde{v}}$, to a larger tree $S_{\tilde{v}}$. **Bottom left:** Recursive computation of the $\mathfrak{o}_{\tilde{v}}^v$. **Bottom middle and right:** Recursive computation of the $\mathfrak{d}_r^v$ in a rooted tree as in Algorithm 2, and of the $\mathfrak{d}_{\tilde{v}}^v$ on a DAG $G_{\tilde{v}}$ as in Algorithm 3.

where $M(v)$ is a $\delta \times \delta$ matrix whose entry $[M(v)]_{ij}$ counts how many times $v$ appears at the $i^{th}$ coordinate of a shortest path in $G$ of discrete length $j$, and $\delta = \max\{\delta(G), \delta(G')\}$. More precisely,

$$
\begin{aligned}
[M(v)]_{ij} &= \text{number of times } v \text{ appears as the } i^{th} \text{ node on a shortest path of discrete length } j \\
&= \textstyle\sum_{\tilde{v} \in V} \text{ number of times } v \text{ appears as } i^{th} \text{ node on a shortest path from } \tilde{v} \\
&\quad \text{ of discrete length } j \\
&= \textstyle\sum_{\tilde{v} \in V} \mathfrak{D}_{\tilde{v}}(v, j - i + 1)\mathfrak{O}_{\tilde{v}}(v, i).
\end{aligned}
$$
(6)

Here $\mathfrak{D}_{\tilde{v}}$ is a $n \times \delta$ matrix whose $(v, i)$-coordinate counts the number of directed walks with $i$ nodes starting at $v$ in the shortest path DAG $G_{\tilde{v}}$. The $\mathfrak{O}_{\tilde{v}}$ is a $n \times \delta$ matrix whose $(v, i)$-coordinate counts the number of directed walks from $\tilde{v}$ to $v$ in $G_{\tilde{v}}$ with $i$ nodes. Given the matrices $\mathfrak{D}_{\tilde{v}}$ and $\mathfrak{O}_{\tilde{v}}$, we compute all $M(v)$ by looping through all choices of source node $\tilde{v} \in V$, adding up the contributions $M_{\tilde{v}}$ to $M(v)$ from each $\tilde{v}$, as detailed in Algorithm 4.

The $v^{th}$ row of $\mathfrak{O}_{\tilde{v}}$, denoted $\mathfrak{o}_{\tilde{v}}^v$, is computed recursively by message-passing from the root, as detailed in Figure 1 and Algorithm 1. Here, $V_{\tilde{v}}^j$ consists of the nodes $v \in V$ for which the shortest paths $\pi_{\tilde{v}v}$ of highest discrete length have $j$ nodes. Algorithm 1 sends one message of size at most $\delta$ per edge, thus has complexity $\mathcal{O}(m\delta)$.

To compute the $v^{th}$ row of $\mathfrak{D}_{\tilde{v}}$, denoted $\mathfrak{d}_{\tilde{v}}^v$, we draw inspiration from [19] where the vectors $\mathfrak{d}_{\tilde{v}}^v$ are computed easily for trees using a message-passing algorithm as follows. Let $T = (V, E, r)$ be a tree with a designated root node $r$. The $i^{th}$ coefficient of $\mathfrak{d}_r^v$ counts the number of paths from $v$ in $T$ of discrete length $i$, directed from the root. This is just the number of descendants of $v$ at level $i$ below $v$ in $T$. Let $\oplus$ denote left aligned addition of vectors of possibly different length, e.g.

$$[a, b, c] \oplus [d, e] = [(a + d), (b + e), c].$$
(7)

Using $\oplus$, the $\mathfrak{d}_r^v$ can be expressed recursively:

$$\mathfrak{d}_r^v = [1] \bigoplus_{p(w)=v} [0, \mathfrak{d}_r^w].$$

---

**Algorithm 1** Message-passing algorithm for computing $\mathfrak{o}_{\tilde{v}}^v$ for all $v$, on $G_{\tilde{v}}$

---

1: Initialize: $\mathfrak{o}_{\tilde{v}}^{\tilde{v}} = [1]$; $\mathfrak{o}_{\tilde{v}}^v = [0] \; \forall \, v \in V \setminus \{\tilde{v}\}$.
2: **for** $j = 1 \ldots \delta$ **do**
3:     **for** $v \in V_{\tilde{v}}^j$ **do**
4:         **for** $(v, w) \in E_{\tilde{v}}$ **do**
5:             $\mathfrak{o}_{\tilde{v}}^w = \mathfrak{o}_{\tilde{v}}^w \oplus [0, \mathfrak{o}_{\tilde{v}}^v]$
6:         **end for**
7:     **end for**
8: **end for**

---

---

**Algorithm 2** Recursive computation of $\mathfrak{d}_r^v$ for all $v$ on $T = (V, E, r)$.

---
1: Initialize: $\mathfrak{d}_r^v = [1] \; \forall \; v \in V$.
2: **for** $e = (v, c(v)) \in E$ **do**
3:    $\mathfrak{d}_r^v = \mathfrak{d}_r^v \oplus [0, \mathfrak{d}_r^{c(v)}]$
4: **end for**

---

---

**Algorithm 3** Recursive computation of $\mathfrak{d}_{\tilde{v}}^v$ for all $v$ on $G_{\tilde{v}}$

---
1: Initialize: $\mathfrak{d}_{\tilde{v}}^v = [1] \; \forall \; v \in V$.
2: **for** $e = (v, c(v)) \in E_G$ **do**
3:    $\mathfrak{d}_{\tilde{v}}^v = \mathfrak{d}_{\tilde{v}}^v \oplus [0, \mathfrak{d}_{\tilde{v}}^{c(v)}]$
4: **end for**

---

The $\mathfrak{d}_r^v$ for all $v \in V$ are computed recursively, sending counters along the edges from the leaf nodes towards the root, recording the number of descendants of any node at any level, see Algorithm 2 and Figure 1. The $\mathfrak{d}_r^v$ for all $v \in V$ are computed in $\mathcal{O}(nh)$ time, where $h$ is tree height, since each edge passes exactly one message of size $\leq h$.

On a DAG, computing $\mathfrak{d}_{\tilde{v}}^v$ is a little more complex. Note that the DAG $G_{\tilde{v}}$ generated by all shortest paths from $\tilde{v} \in V$ can be expanded into a rooted tree $S_{\tilde{v}}$ by duplicating any node with several incoming edges, see Figure 1. The tree $S_{\tilde{v}}$ contains, as a path from the root $\tilde{v}$ to one of the nodes labeled $v$ in $S_{\tilde{v}}$, any shortest path from $\tilde{v}$ to $v$ in $G$. However, the number of nodes in $S_{\tilde{v}}$ could, in theory, be exponential in $n$, making computation of $\mathfrak{d}_{\tilde{v}}^v$ by message-passing on $S_{\tilde{v}}$ intractable. Thus, we shall compute the $\mathfrak{d}_{\tilde{v}}^v$ on the DAG $G_{\tilde{v}}$ rather than on $S_{\tilde{v}}$. As on trees, the $\mathfrak{d}_{\tilde{v}}^v$ in $S_{\tilde{v}}$ are given by $\mathfrak{d}_{\tilde{v}}^v = [1] \oplus \bigoplus_{(w,v) \in E_{\tilde{v}}} [0, \mathfrak{d}_{\tilde{v}}^w]$, where $\oplus$ is defined in (7). This observation leads to an algorithm in which each edge $e \in E_{\tilde{v}}$ passes exactly one vector of size $\leq \delta + 1$ in the direction of the root $\tilde{v}$, starting at the leaves of the DAG $G_{\tilde{v}}$ and computing updated descendant vectors for each receiving node. See Algorithm 3 and Figure 1. The complexity of Algorithm 3, which computes $\mathfrak{d}_{\tilde{v}}^v$ for all $v \in V$, is $\mathcal{O}(|E_{\tilde{v}}|\delta) \leq \mathcal{O}(m\delta)$.

### 2.2 Computational complexity analysis

Given the $w(v, v')$ and the $k_n(v, v')$ for all $v \in V$ and $v' \in V'$, the kernel can be computed in $\mathcal{O}(n^2)$ time. If we assume that each node kernel $k_n(v, v')$ can be computed in $\mathcal{O}(d)$ time (as is the case with many standard kernels including Gaussian and linear kernels), then all $k_n(v, v')$ can be precomputed in $\mathcal{O}(n^2 d)$ time. Given the matrices $M(v)$ and $M(v')$ for all $v \in V$, $v' \in V'$, each $w(v, v')$ requires $\mathcal{O}(\delta^2)$ time, giving $\mathcal{O}(n^2 \delta^2)$ complexity for computing all weights $w(v, v')$.

Note that Algorithm 4 computes $M(v)$ for all $v \in G$ simultaneously. Adding the time complexities of the lines in each iteration of the algorithm as given on the right hand side of the individual lines in Algorithm 4, the total complexity of one iteration of Algorithm 4 is

$$\mathcal{O}\left((mn + n\log n) + m\delta + m\delta + n\delta^2 + n\delta^2\right) = \mathcal{O}(n(m + \log n + \delta^2)),$$

---

**Algorithm 4** Algorithm simultaneously computing all $M(v)$

---
1: Initialize: $M(v) = \mathbf{0} \in \mathbb{R}^{\delta \times \delta}$ for each $v \in V$.
2: **for all** $\tilde{v} \in V$ **do**
3:    compute shortest path DAG $G_{\tilde{v}}$ rooted at $\tilde{v}$ using Dijkstra                     $(\mathcal{O}(mn + n\log n))$
4:    compute $\mathfrak{D}_{\tilde{v}}(v)$ for each $v \in V$                                           $(\mathcal{O}(m\delta))$
5:    compute $\mathfrak{O}_{\tilde{v}}(v)$ for each $v \in V$                                           $(\mathcal{O}(m\delta))$
6:    for each $v \in V$, compute the $\delta \times \delta$ matrix $M_{\tilde{v}}(v)$ given by

$$[M_{\tilde{v}}(v)]_{ij} = \begin{cases} \mathfrak{D}_{\tilde{v}}(v, j - i + 1)\mathfrak{O}_{\tilde{v}}(v, i) & \text{when } i \leq j \\ 0 & \text{otherwise,} \end{cases} \qquad (\mathcal{O}(n\delta^2))$$

7:    update $M(v) = M(v) + M_{\tilde{v}}(v)$ for each $v \in V$                             $(\mathcal{O}(n\delta^2))$
8: **end for**

---

giving total complexity $\mathcal{O}(n^2(m+\log n+\delta^2))$ for computing $M(v)$ for all $v \in V$ using Algorithm 4. It follows that the total complexity of computing $k(G, G')$ is

$$\mathcal{O}(n^2 + n^2 d + n^2\delta^2 + n^2\delta^2 + n^2(m + \log n + \delta^2)) = \mathcal{O}(n^2(m + \log n + d + \delta^2)).$$

When computing the kernel matrix $K_{ij} = k(G_i, G_j)$ for a set $\{G_i\}_{i=1}^N$ of graphs with $N > m + n + \delta^2$, note that Algorithm 4 only needs to be run once for every graph $G_i$. Thus, the average complexity of computing one kernel value out of all $K_{ij}$ becomes

$$\frac{1}{N^2}\left(N\mathcal{O}(n^2(m + \log n + \delta^2)) + N^2\mathcal{O}(n^2 + n^2 d + \delta^2)\right) \leq \mathcal{O}(n^2 d).$$

## 3 Experiments

Classification experiments were made with the proposed GraphHopper kernel and several alternatives: The propagation kernel PROP [16], the connected subgraph matching kernel CSM [14] and the shortest path kernel SP [17] all use continuous-valued attributes. In addition, we benchmark against the Weisfeiler-Lehman kernel WL [2], which only uses discrete node attributes. All kernels were implemented in Matlab, except for CSM, where a Java implementation was supplied by N. Kriege. For the WL kernel, the Matlab implementation available from [20] was used. For the GraphHopper and SP kernels, shortest paths were computed using the BGL package [21] implemented in C++. The PROP kernel was implemented in two different versions, both using the total variation hash function, as the Hellinger distance is only directly applicable to positive vector-valued attributes. For PROP-diff, labels were propagated with the diffusion scheme, whereas in PROP-WL labels were first discretised via hashing and then the WL kernel [2] update was used. The bin width of the hash function was set to $10^{-5}$ as suggested in [16]. The PROP-diff, PROP-WL and the WL kernel were each run with 10 iterations. In the CSM kernel, the clique size parameter was set to $k = 5$. Our kernel implementations and datasets (with the exception of AIRWAYS) can be found at http://image.diku.dk/aasa/software.php.

Classification experiments were made on four datasets: ENZYMES, PROTEINS, AIRWAYS and SYNTHETIC. ENZYMES and PROTEINS are sets of proteins from the BRENDA database [22] and the dataset of Dobson and Doig [23], respectively. Proteins are represented by graphs as follows. Nodes represent secondary structure elements (SSEs), which are connected whenever they are neighbors either in the amino acid sequence or in $3D$ space [24]. Each node has a discrete type attribute (helix, sheet or turn) and an attribute vector containing physical and chemical measurements including length of the SSE in Ångstrøm (Å), distance between the $C_\alpha$ atom of its first and last residue in Å, its hydrophobicity, van der Waals volume, polarity and polarizability. ENZYMES comes with the task of classifying the enzymes to one out of 6 EC top-level classes, whereas PROTEINS comes with the task of classifying into enzymes and non-enzymes. AIRWAYS is a set of airway trees extracted from CT scans of human lungs [25, 26]. Each node represents an airway branch, attributed with its length. Edges represent adjacencies between airway bronchi. AIRWAYS comes with the task of classifying airways into healthy individuals and patients suffering from Chronic Obstructive Pulmonary Disease (COPD). SYNTHETIC is a set of synthetic graphs based on a random graph $G$ with 100 nodes and 196 edges, whose nodes are endowed with normally distributed scalar attributes sampled from $\mathcal{N}(0, 1)$. Two classes $A$ and $B$ each with 150 attributed graphs were generated from $G$ by randomly rewiring edges and permuting node attributes. Each graph in $A$ was generated by rewiring 5 edges and permuting 10 node attributes, and each graph in $B$ was generated by rewiring 10 edges and permuting 5 node attributes, after which noise from $\mathcal{N}(0, 0.45^2)$ was added to every node attribute in every graph. Detailed metrics of the datasets are found in Table 1.

Both GraphHopper, SP and CSM depend on freely selected node kernels for continuous attributes, giving modeling flexibility. For the ENZYMES, AIRWAYS and SYNTHETIC datasets, a Gaussian node kernel $k_n(v, v') = e^{-\lambda\|A(v)-A(v')\|^2}$ was used on the continuous-valued attribute, with $\lambda = 1/d$. For the PROTEINS dataset, the node kernel was a product of a Gaussian kernel with $\lambda = 1/d$ and a Dirac kernel on the continuous- and discrete-valued node attributes, respectively. For the WL kernel, discrete node labels were used when available (in ENZYMES and PROTEINS); otherwise node degree was used as node label.

Classification was done using a support vector machine (SVM) [27]. The SVM slack parameter was trained using nested cross validation on 90% of the entire dataset, and the classifier was tested on the

|                          | ENZYMES        | PROTEINS | AIRWAYS | SYNTHETIC |
|--------------------------|----------------|----------|---------|-----------|
| Number of nodes          | 32.6           | 39.1     | 221     | 100       |
| Number of edges          | 46.7           | 72.8     | 220     | 196       |
| Graph diameter           | 12.8           | 11.6     | 21.1    | 7         |
| Node attribute dimension | 18             | 1        | 1       | 1         |
| Dataset size             | 600            | 1113     | 1966    | 300       |
| Class size               | $6 \times 100$ | 663/450  | 980/986 | 150/150   |

Table 1: Data statistics: Average node and edge counts and graph diameter, dataset and class sizes.

| Kernel        | ENZYMES                          | PROTEINS                    | AIRWAYS                     | SYNTHETIC                          |
|---------------|----------------------------------|-----------------------------|-----------------------------|------------------------------------|
| GraphHopper   | $\mathbf{69.6 \pm 1.3}$ (12'10'')| $74.1 \pm 0.5$ (2.8 h)      | $\mathbf{66.8 \pm 0.5}$ (1 d 7 h) | $\mathbf{86.6 \pm 1.0}$ (12'10'') |
| PROP-diff [16]| $37.2 \pm 2.2$ (13'')            | $73.3 \pm 0.4$ (26'')       | $63.5 \pm 0.5$ (4'12'')     | $46.1 \pm 1.9$ (1'21'')            |
| PROP-WL [16]  | $48.5 \pm 1.3$ (1'9'')           | $73.1 \pm 0.8$ (2'40'')     | $61.5 \pm 0.6$ (8'17'')     | $44.5 \pm 1.2$ (1'52'')            |
| SP [17]       | $\mathbf{71.0 \pm 1.3}$ (3 d)    | $\mathbf{75.5 \pm 0.8}$ (7.7 d) | OUT OF TIME             | $\mathbf{85.4 \pm 2.1}$ (3.4 d)    |
| CSM [14]      | $\mathbf{69.4 \pm 0.8}$          | OUT OF MEMORY               | OUT OF MEMORY               | OUT OF TIME                        |
| WL [2]        | $48.0 \pm 0.9$ (18'')            | $\mathbf{75.6 \pm 0.5}$ (2'51'') | $62.0 \pm 0.6$ (7'43'')| $43.3 \pm 2.3$ (2'8'')             |

Table 2: Mean classification accuracies with standard deviation for all experiments, significantly best accuracies in bold. OUT OF MEMORY means that 100 GB memory was not enough. OUT OF TIME indicates that the kernel computation did not finish within 30 days. Runtimes are given in parentheses; see Section 3.1 for further runtime studies. Above, $x'y''$ means $x$ minutes, $y$ seconds.

remaining 10%. This experiment was repeated 10 times. Mean accuracies with standard deviations are reported in Table 2. For each kernel and dataset, runtime is given in parentheses in Table 2. Runtimes for the CSM kernel are not included, as this implementation was in another language.

## 3.1   Runtime experiments

An empirical evaluation of the runtime dependence on the parameters $n$, $m$ and $\delta$ is found in Figure 2. In the top left panel, average kernel evaluation runtime was measured on datasets of 10 random graphs with $10, 20, 30, \ldots, 500$ nodes each, and a density of $0.4$. Density is defined as $\frac{m}{n(n-1)/2}$, i.e. the fraction of edges in the graph compared to the number of edges in the complete graph. In the top right panel, the number of nodes was kept constant $n = 100$, while datasets of 10 random graphs were generated with $110, 120, \ldots, 500$ edges each. Development of both average kernel evaluation runtime and graph diameter is shown. In the bottom panels, the relationship between runtime and graph diameter is shown on subsets of 100 and 200 of the real AIRWAYS and PROTEINS datasets, respectively, for each diameter.

## 3.2   Results and discussion

Our experiments on ENZYMES and AIRWAYS clearly demonstrate that there are real-world classification problems where continuous-valued attributes make a big contribution to classification performance. Our experiments on SYNTHETIC demonstrate how the more discrete types of kernels, PROP and WL, are unable to classify the graphs. Already on SYNTHETIC, which is a modest-sized set of modest-sized graphs, CSM and SP are too computationally demanding to be practical, and on AIRWAYS, which is a larger set of larger trees, they cannot finish in 30 days. The CSM kernel [14] has asymptotic runtime $\mathcal{O}(kn^{k+1})$, where $k$ is a parameter bounding the size of subgraphs considered by the kernel, and thus in order to study subgraphs of relevant size, its runtime will be at least as high as the shortest path kernel. Moreover, the CSM kernel requires the computation of a product graph which, for graphs with hundreds of nodes, can cause memory problems, which we also find in our experiments. The PROP kernel is fast; however, the reason for the computational efficiency of PROP is that it is not really a kernel for continuous valued features – it is a kernel for discrete features combined with a hashing scheme to discretize continuous-valued features. In our experiments, these hashing schemes do not prove powerful enough to compete in classification accuracy with the kernels that really do use the continuous-valued features.

While ENZYMES and AIRWAYS benefit significantly from including continuous attributes, our experiments on PROTEINS demonstrate that there are also classification problems where the most important information is just as well summarized in a discrete feature: here our combination of

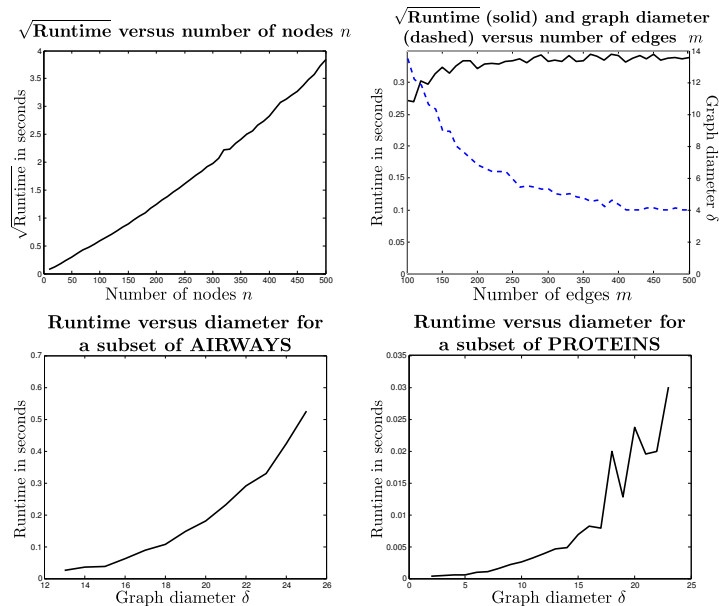

Figure 2: Dependence of runtime on $n$, $\delta$ and $m$ on synthetic and real graph datasets.

continuous and discrete node features gives equal classification performance as the more efficient WL kernel using only discrete attributes.

We proved in Section 3.1 that the GraphHopper kernel has asymptotic runtime $\mathcal{O}(n^2(d+m+\log n+\delta^2))$, and that the average runtime for one kernel evaluation in a Gram matrix is $\mathcal{O}(n^2d)$ when the number of graphs exceeds $m+n+\delta^2$. Our experiments in Section 3.1 empirically demonstrate how runtime depends on the parameters $n$, $m$ and $\delta$. As $m$ and $\delta$ are dependent parameters, the runtime dependence on $m$ and $\delta$ is not straightforward. An increase in the number of edges $m$ typically leads to an increased graph diameter $\delta$ for small $m$, but for more densely connected graphs, $\delta$ will decrease with increasing $m$ as seen in the top right panel of Figure 2. A consequence of this is that graph diameter rarely becomes very large compared to $m$. The same plot also shows that the runtime increases slowly with increasing $m$. Our runtime experiments clearly illustrate that while in the worst case scenario we could have $m = n^2$ or $\delta = n$, this rarely happens in real-world graphs, which are often sparse and with small diameter. Our experiments also illustrate an average runtime quadratic in $n$ on large datasets, as expected based on complexity analysis.

## 4   Conclusion

We have defined the GraphHopper kernel for graphs with any type of node attributes, presented an efficient algorithm for computing it, and demonstrated that it outperforms state-of-the-art graph kernels on real and synthetic data in terms of classification accuracy and/or speed. The kernels are able to take advantage of any kind of node attributes, as they can integrate any user-defined node kernel. Moreover, the kernel is parameter-free except for the node kernels.

This kernel opens the door to new application domains such as computer vision or medical imaging, in which kernels that work solely on graphs with discrete attributes were too restrictive so far.

## Acknowledgements

The authors wish to thank Nils Kriege for sharing his code for computing the CSM kernel, Nino Shervashidze and Chloé-Agathe Azencott for sharing their preprocessed chemoinformatics data, and Asger Dirksen and Jesper Pedersen for sharing the AIRWAYS dataset. This work is supported by the Danish Research Council for Independent Research | Technology and Production, the Knud Høygaard Foundation, AstraZeneca, The Danish Council for Strategic Research, Netherlands Organisation for Scientic Research, and the DFG project "Kernels for Large, Labeled Graphs (LaLa)". The research of Professor Dr. Karsten Borgwardt was supported by the Alfried Krupp Prize for Young University Teachers of the Alfried Krupp von Bohlen und Halbach-Stiftung.

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
