[Reviews · NeurIPS 2013]

Submitted by Assigned_Reviewer_4

This paper presents scalable graph kernels for graphs with continuous attributes, which computes kernels based on subpaths shared between two graphs. The computation time is quadratic for the number of nodes in graphs, which is smaller than cubic of existing subpath kernels. Experimental results using benchmark datasets show that higher accuracies of the proposed method than those of state-of-the-art methods and practical computational times. The paper is well written and presents a nice idea to compute kernels for graphs with continuous attributes.
Summary: The proposed methods is similar to PROP in that both methods propagate label propagations for kernel computations. It would be better that the authors discuss the major differences of two methods and why the proposed method achieves higher accuracies.

Submitted by Assigned_Reviewer_5

This paper proposed a new kernel function and its fast computation algorithm for graphs with continuous attribute values. The proposed algorithm is much more efficient than competitive methods. The advantages of proposed method was proved by theoretically and evaluated by numerical experiments.

Quality:
The efficiency of the proposed method was through investigated both by theory (proved the computational complexity) and by numerical experiments, which shows the proposed method is much better than related works. But the new kernel function itself was not analyzed enough, which was only evaluated by performance results on classification tasks. One might have the question which pairs of graphs are similar/dissimilar on the proposed kernel (similarity measure). The comparison with other kernel functions using toy graph examples would be necessary.

Clarity:
The paper is well organized and understandable for readers.

Originality:
This paper proposed a novel method, with a new kernel for graphs and furthermore proposed a fast computation algorithm for the new kernel. The proposed method can manage massive datasets of graphs with continuous attribute values, which cannot be managed by conventional methods.

Significance:
The proposed method and results would be interesting. Because the result is general, this method could be applicable in a lot of applications.


Other comments:
1.
The proposed computation might evaluate the kernel values for same pairs of paths multiple times, which means that the algorithm might assume the path from $v_a$ to $v_b$ and the path from $v_b$ to $v_a$ as different paths. If it is true, analyzing how it could affect to the similarity measure would be important.

2.
The caption in Table 2 should mention the meaning of the real values even if it is mentioned in the main sentence. And, what are the values in the parentheses in the row of SP? If it is average runtimes, they are less than 24 hours. Then the runtime of SP should not be OUT OF TIME.

Summary: This paper proposed a novel graph kernel function and a scalable computation algorithm. The result would be interesting for researchers in machine leaning and application areas.

Submitted by Assigned_Reviewer_6

The paper proposes an algorithm for computing a graph kernel based on shortest-paths, with the specific goal of handling continuous valued attributes in the nodes. I agree with the authors that this goal is worth investigation.

The kernel itself is just one possible variant of the shortest-path kernel in [19], so there is no substantial novelty on this side. The contribution of this paper is an algorithm for computing this kernel, which is sound and asymptotically faster than the method in [19], as also confirmed by the experiments. Still, I'm not convinced that this paper brings a significant advancement to the state-of-the-art. Other (and possibly much faster) graph kernels exist and the comparisons against those is unsatisfactory. In particular: What happens of the continuous attributes when using the WL kernel? The author say that it ".. only uses discrete node attributes." but this is not a fair comparison since the continuous attributes can always be discretized and such additional information could significantly reduce the accuracy gap. Another possible (very fast) graph kernel is the NSPKD by Costa & De Grave (2010), which is also designed for categorical attributes, but again discretization could be used.
Summary: Reasonably well organized paper trying to bring kernels to work in a useful context. Unclear that it really advances the state-of-the-art.
Author Feedback

Author rebuttal: We thank the reviewers for their positive feedback and valuable comments on our paper, which introduces a fast and novel kernel for graphs with continuous attributes.

We would like to correct a couple of misunderstandings by Reviewers 4 and 6 and answer questions by Reviewer 5.

REVIEWER 4:
* Our kernel is not a case of, or even similar to, the propagation kernel PROP [18]. Our message-passing algorithm computes node kernel weights which rely on the global topology of the two graphs, and our kernel uses the actual continuous node attributes. PROP compares propagation of local configurations of discretized attributes [18]. We will further stress this difference in the revised paper.

REVIEWER 6:
* Reviewer 6 requests comparison with a WL-type kernel on discretized attributes, which we do, in fact, have: namely the PROP kernel [18]. We chose this instance representing the family of WL type kernels (PROP, WL, Costa-De Grave) on discretized attributes, as it achieved the best results in initial classification experiments. Still, it is outperformed by our novel kernel.
* More generally, discretization is not the solution to continuous attributes, which can potentially be very high dimensional. This is supported by our experiments comparing to PROP.
* The proposed kernel is *not* a special case of the SP kernel, as the kernel on paths is different: The SP kernel compares shortest path lengths in the two graphs; our kernel compares all nodes along the path, hence is geometry-aware while the SP kernel is not. The difference in path kernel is key to the improved runtime and scalability).

ANSWERS TO REVIEWER 5:
1. The path kernel discriminates the paths pi_ab and pi_ba, but note: the graph kernel sums over all paths. For corresponding vertex pairs a, b in G and G', the path pairs (pi_ab, pi'_ab) and (pi_ba, pi'_ba) contribute strongly to the kernel value k(G,G').
2. The parantheses in table 2 contain average SP runtime for a single k(G,G'), giving roughly estimated dataset runtimes of 1/30/19674/15 days. This is why the classification ran out of time for all but the first.